# Enantioseparation of Proton Pump Inhibitors by HPLC on Polysaccharide-Type Stationary Phases: Enantiomer Elution Order Reversal, Thermodynamic Characterization, and Hysteretic Effect

**DOI:** 10.3390/ijms26157217

**Published:** 2025-07-25

**Authors:** Máté Dobó, Gergely Molnár, Ali Mhammad, Gergely Dombi, Arash Mirzahosseini, Zoltán-István Szabó, Gergő Tóth

**Affiliations:** 1Department of Pharmaceutical Chemistry, Semmelweis University, Hőgyes E. Str. 9, H-1092 Budapest, Hungary; dobo.mate@stud.semmelweis.hu (M.D.); molnar.gergely@stud.semmelweis.hu (G.M.); ali.mhammad@phd.semmelweis.hu (A.M.); dombi.gergely@semmelweis.hu (G.D.); mirzahosseini.arash@semmelweis.hu (A.M.); 2Center for Pharmacology and Drug Research & Development, Semmelweis University, H-1092 Budapest, Hungary; 3Department of Pharmaceutical Industry and Management, George Emil Palade University of Medicine, Pharmacy, Science and Technology of Targu Mures, Gh. Marinescu 38, 540139 Targu Mures, Romania; 4Sz-Imfidum Ltd., Lunga nr. 504, 525401 Lunga, Romania

**Keywords:** chiral separation, enantioseparation, chiral sulfoxides, proton pump inhibitor, esomeprazole, hysteresis, polysaccharide type chiral column

## Abstract

The separation of three proton pump inhibitors (omeprazole, lansoprazole, and rabeprazole) as exemplified molecules containing chiral sulfoxide groups was investigated in polar organic liquid chromatographic mode on seven different polysaccharide stationary phases (Chiralcel OD and OJ; Chiralpak AD, AS, and IA; Lux Cellulose-2 and -4). Different alcohols, such as methanol, ethanol, 1-propanol, 2-propanol, and their combinations, were used as eluents. After method optimization, semi-preparative enantioseparation was successfully applied for the three proton pump inhibitors to collect the individual enantiomers. A detailed investigation was conducted into elution order reversal, thermodynamic parameters, the effect of eluent mixtures, and the hysteresis of retention time and selectivity. Using Chiralpak AS, containing the amylose tris[(S)-α-methylbenzylcarbamate] chiral selector, the separation of the investigated enantiomers was achieved in all four neat eluents, with methanol providing the best results. In many cases, a reversal of the enantiomer elution order was observed. In addition to chiral-selector-dependent reversal, eluent-dependent reversal was also observed. Notably, even replacing methanol with ethanol altered the enantiomer elution order. Both enthalpy- and entropy-controlled enantioseparation were also observed in several cases; however, temperature-dependent elution order reversal was not. The hysteresis of retention and selectivity was further investigated on amylose-type columns in methanol–2-propanol and methanol–ethanol eluent mixtures. The phenomenon was observed on all amylose columns regardless of the eluent mixtures employed. Hystereticity ratios were calculated and used to compare the hysteresis behaviors of different systems. Multivariate statistical analysis revealed that Chiralpak AS exhibited the most distinct enantioselective behavior among the tested columns, likely due to the absence of a direct connection between the carbamate moiety and the aromatic substituent. The present study aided in understanding the mechanisms leading to enantiomer recognition, which is crucial for developing new chiral stationary phases and chiral HPLC method development in general.

## 1. Introduction

Exploring enantioseparations often involves a more elaborate developmental trajectory compared to achiral separations. A thorough grasp of the molecular mechanisms governing enantioseparation is indispensable for the systematic advancement of chiral separation methodologies. Consequently, a comprehensive examination of these mechanisms becomes pivotal in augmenting predictive capacities. One way to gain deeper insights into enantiorecognition involves investigating structurally related compounds under varied conditions, wherein parameters undergo systematic alterations. This approach provides a nuanced understanding of how changes in conditions impact enantioselectivity, thus contributing to a more comprehensive understanding of enantioseparation mechanisms [1,2,3,4,5,6,7,8]. In the present study, proton pump inhibitors (PPIs) with a chiral sulfoxide group were selected as exemplified molecules.

Chiral sulfur and phosphorus-containing molecules play crucial roles in organic chemistry and agrochemicals, contributing to their significance in various fields. Despite the absence of approved pharmaceutical agents containing chiral phosphorus, the chiral sulfoxide group, which comprises PPIs, is among the most frequently prescribed medications. These medications are frequently used for the treatment of conditions such as gastroesophageal reflux disease, peptic ulcers, and other acid-related disorders [9]. The first PPI introduced for clinical use was omeprazole. It was developed by Astra AB (now AstraZeneca, Gaithersburg, MD, USA) and was first marketed under the brand name Losec in the late 1980s. Omeprazole revolutionized the treatment of acid-related disorders, offering a more potent and specific mechanism of action compared to previous acid-suppressing medications like histamine H2-receptor antagonists (e.g., cimetidine, ranitidine) [10]. Since the introduction of omeprazole, several other PPIs with similar moiety, such as lansoprazole, pantoprazole and rabeprazole, have been developed and are widely used in clinical practice. All these drugs contain a chiral sulfoxide group, and therefore, they can exist in two enantiomeric forms. The structure of the compounds investigated can be found in Figure 1. Initially, PPIs were marketed as racemates containing both enantiomers. However, as research into their pharmacokinetics and stereoselective metabolism advanced, it became evident that one enantiomer was often more effective. This led to the development of new formulations containing only the more effective enantiomers. Consequently, formulations containing esomeprazole, dexlansoprazole, or dexrabeprazole emerged because of these chiral switches, offering improved therapeutic outcomes for patients with acid-related disorders [11,12]. To investigate the enantiomeric purity and stereoselective behavior of PPIs, chiral analytical methods are necessary. Among various enantioselective methods, HPLC on a chiral stationary phase (CSP) is the gold standard in this field [13,14,15]. There are now hundreds of CSPs available on the market; however, polysaccharide-type stationary phases with an amylose or cellulose backbone are the most widely used because of their high enantioseparation capacity and multimodal nature. These columns can be used in normal-phased, reversed-phased, and polar organic (PO) modes as well. Because the normal-phased mode uses toxic, harmful eluents, generally, its application should be avoided, even for preparative purposes. However, the normal-phase mode can be usually substituted by the PO mode, which presents several advantages when compared to the normal-phased mode. The PO mode uses only alcoholic eluents (methanol (MeOH), ethanol (EtOH), 1-propanol (1-PrOH), and 2-propanol (2-PrOH)), acetonitrile, or the mixtures thereof. Advantages such as shorter run times, heightened efficiency, simplified coupling with mass spectrometry, and the increased solubility of analytes in the mobile phase [2,16,17] make this mode especially attractive for chiral method development. Furthermore, the utilization of the PO mode has proven to be advantageous for both analytical and (semi-)preparative purposes [18,19]. Several studies have explored the enantioseparation of PPIs, employing techniques such as capillary electrophoresis [20], SFC [21], and HPLC [22,23]. The previous literature methods were also summarized in a recent review [24]. Nevertheless, the literature notably lacks systematic measurements and a semi-preparative enantioselective method in the PO mode, which could offer significant methodological insights and contribute to a deeper understanding of the mechanisms underlying enantioseparation. Nowadays, one of the most interesting topics in POs is the phenomenon of hysteresis of retention and enantioselectivity, first described by Németh and Horváth on an amylose tris(3,5-dimethylphenylcarbamate) column using MeOH:2-PrOH mixtures [25]. Later, it became clear that hysteresis is not a unique phenomenon and can be generally observed in various eluent mixtures on amylose-based columns. Moreover, a recent study from the Ilisz group shows that cellulose columns can also exhibit hysteresis, although to a lesser extent compared to amylose-based columns [26].

This study aimed to systematically investigate the chiral separation of three PPIs—omeprazole, lansoprazole, and rabeprazole (Figure 1)—in PO mode. It seeks to understand the enantiorecognition capacity of a chiral polysaccharide column by examining the influence of different alcohols as an eluent. Additionally, this study also aimed to delve deeper into the phenomenon of enantiomeric elution order (EEO) reversal, analyze the thermodynamic parameters of separation including multivariate statistical analysis, and scrutinize hysteresis effects on amylose-type columns. As a practical application, a semi-preparative method was also developed to collect enantiomers from each racemate.

## 2. Results and Discussions

### 2.1. Enantioseparation Screening in Pure Eluents

As a first step in enantioseparation, a screening study of four cellulose (Chiralcel OJ, OD, Lux Cellulose-2, Lux Cellulose-4) and three amylose-based (Chiralpak AD, Chiralpak AS, Chiralpak IA) CSPs was conducted using neat eluents in PO mode, namely MeOH, EtOH, 1-PrOH, and 2-PrOH, at 20 °C and using 0.7 mL/min flow rate. Chiralcel OD and Chiralpak AD contain the same substituent; these two columns differ only in the backbone. Chiralpak IA and Chiralpak AD both employ the same amylose tris(3,5-dimethylphenylcarbamate) as the chiral selector. The key difference lies in the mode of attachment of the chiral selector to the silica particles: in the Chiralpak IA column, the chiral selector is immobilized, whereas in the Chiralpak AD column and other chiral columns tested in this work, the chiral selector is applied through a coating process. Lux Cellulose-2 and Lux Cellulose-4 differ only in the position of the substituent in the side chain. The structures of the applied chiral selectors used in this study are depicted in Figure 2.

Out of the 84 measurements conducted (comprising seven stationary phases multiplied by four neat eluents and three PPIs), enantiorecognition was achieved in 59 instances, resulting in a success rate of 70%. The outcomes of this screening are detailed in Table 1, and some representative chromatograms can be seen in Figure 3.

The analysis of retention factors indicates that the three enantiomeric pairs exhibited similar behavior under identical chromatographic conditions. When using cellulose-based columns, the retention factor generally increased following this order: MeOH < EtOH < 1-PrOH < 2-PrOH, with a few exceptions. Interestingly, this trend was not observed on amylose-based columns. For example, on the Chiralpak AD column, the lowest retention factor was observed using 2-PrOH, and the highest was measured using neat EtOH as the mobile phase (Appendix A). It was interesting to observe that the amylose-based Chiralpak AS column behaved similarly to the cellulose-based CSPs. When omeprazole and rabeprazole were investigated, MeOH provided the lowest retention times and 2-PrOH the highest. The highest retention factor was observed with the chlorinated cellulose columns, such as Lux Cellulose-2 and Lux Cellulose-4. This indicates that electron-withdrawing groups, like halogens, play a role in binding to the stationary phase, regardless of their substitution position.

The screening data was further evaluated by analyzing the resolution values (R_s_). The 70% success rate and the high number of baseline separation (R_s_ > 1.5) (38.5%) showed that PO mode employing alcohols on polysaccharide columns is a versatile method for the enantioseparation of PPIs. Interestingly, Chiralcel OJ displayed no enantiorecognition for these compounds; however, at least one compound could be baseline-separated with all the other columns. Chiralcel OJ was the only column with a benzoate-type selector whereas all other columns possessed a carbamate linker. Applying Lux Cellulose-2 with MeOH and Lux Cellulose-4 with MeOH and EtOH baseline separation was achieved for all enantiomeric pairs. The sums of the R_s_ values for the CSP–mobile phase pairs and the CSP–analyte pairs were studied in more detail. These data are depicted in Figure 4.

The highest sum R_s_ value was observed on the Lux Cellulose-4 CSP. Among the eluents, MeOH provided the best results, achieving the highest values. The best and the worst column were also cellulose-based, which very well demonstrated the importance of the type of substituent on the chiral selector. Among the amylose-type columns, the immobilized Chiralpak IA performed the best.

The cellulose-based CSPs showed a decrease in the sum Rs in the mobile phase order of MeOH > EtOH > 1-PrOH > 2-PrOH. This correlated with the relative polarity of these alcohols [26]; the increase in relative polarity resulted in the increase in sum Rs. On the contrary, amylose-based CSPs did not show a similar effect. It could be explained by higher-order structure differences because of the different backbones in the chiral selector. Amylose can adopt several distinct stable conformations that can present different enantiorecognition mechanisms, with this state strongly influenced by the eluent used. In contrast, cellulose maintains only one stable conformation, and the utilization of alcohols with longer carbon chains compromises its chiral recognition capability.

It should also be noted that by investigating the sum R_s_ of the CSP–analyte pairs, in most CSPs, the sum was decreasing in the omeprazole > rabeprazole > lansoprazole order. There were two exceptions, Chiralcel OD and Chiralpak AS. Chiralcel OD is characterized by the same 3,5-dimethylphenylcarbamate substituent on the polysaccharide backbone, analogous to that observed in Chiralpak AD. However, significant disparities were observed in the sum R_s_, retention factor, and enantiorecognition by the three analytes. These observations underscore the potential significance of the backbone in these cases.

It is also interesting to compare the Chiralpak IA and Chiralpak AD columns, which contained an identical selector, with the only distinction between them rooted in the way the chiral selector was attached to the silica. The immobilization of the chiral selector in Chiralpak IA resulted in lower retention time compared to Chiralpak AD for all three analytes in most of the eluents tested.

Without further method optimization, the highest R_s_ values were achieved for the three analytes as follows: omeprazole on Chiralpak AD, MeOH (R_s_ = 9.79), rabeprazole on Lux Cellulose-4, MeOH (R_s_ = 6.16), and lansoprazole on Lux Cellulose-4, MeOH (R_s_ = 3.26). Chromatograms with highest resolution values can be seen in Figure 3A–C.

### 2.2. Enantiomer Elution Order Reversal

The reversal of the EEO is crucial as changes in EEO indicate alterations in the enantiorecognition capabilities of a chiral selector. Therefore, studying EEO reversals can provide deeper insights into the design and understanding of chiral separations. According to the literature, changes in any chromatographic parameters can lead to EEO reversal, but they can be broadly categorized into three main types: chiral-selector-dependent EEO reversals, mobile-phase-dependent EEO reversals, and temperature-dependent EEO reversals. In our study, temperature-dependent EEO reversal was not observed; the effect of temperature on enantioseparation is discussed in detail in the following section (Section 2.3). The fact that changing the chiral selector results in the inversion of the enantiomer order is not that surprising. Chiralpak AD and Chiralcel OD columns differ just in the backbone of the selector as they contain the same 3,5-dimethylphenylcarbamate pendant groups. Nonetheless, there are many examples in the literature showing that this change in the structure results in EEO reversal [5,27]. This was also observed in our study in the case of omeprazole in neat 2-PrOH and for rabeprazole using EtOH as the mobile phase. These changes in the EEO indicate that the backbone plays a crucial role in the enantiorecognition mechanism.

EEO reversals due to changes in nature of substituents of the chiral selector are also quite common. Often, EEO reversals occur when the 3,5-dimethylphenylcarbamate substituent is replaced with another type of substituent, regardless of the backbone. Interestingly, on cellulose-type CSPs, omeprazole did not exhibit any substituent-dependent EEO reversal while lansoprazole showed EEO reversal in every investigated eluent when comparing Chiralcel OD to Lux Cellulose-4.

Mobile-phase-dependent EEO reversals were observed only on the amylose-type column, indicating that different types of alcohols could modify the structure of amylose-type CSPs, potentially leading to variations in the EEO. In contrast, such EEO reversals did not occur with cellulose-type columns, suggesting that in these cases, the enantiorecognition mechanism was not significantly altered. The majority of EEO reversals occurred when switching MeOH or EtOH to 1-PrOH or 2-PrOH. Interestingly, EEO reversals frequently occurred when MeOH was replaced with EtOH, highlighting the complexity of the separation system. MeOH-EtOH exchange caused EEO reversal on the Chiralpak AS column for all investigated PPIs: in the case of rabeprazole, using the Chiralpak AD column, and on the Chiralpak IA column, for omeprazole and lansoprazole. Some representative chromatograms regarding EEO reversals are depicted in Appendix A.

### 2.3. Thermodynamic Study of the Enantioseparations

The effect of temperature can be crucial in chiral separations. Thermodynamic calculations are highly valuable and widely used methods for understanding chiral recognition mechanisms in enantioselective HPLC [28]. The purpose of the thermodynamic study in our research was to gain insights into the interaction energies and driving forces governing chiral recognition on different polysaccharide-type columns in PO mode. Measurements were conducted at 10, 20, 25, 30, and 40 °C on six different columns, using all four neat mobile phases. Chiralcel OJ was not investigated because no enantioseparation was observed on this column. The data obtained at different temperatures clearly showed that the retention factor consistently decreased with an increase in temperature, regardless of the analyte being examined. The analysis of selectivity values as a function of temperature showed a more varied picture; in many separation systems, the selectivity was increasing with an increase in a column temperature. The obtained thermodynamic data (Δ(ΔH°), Δ(ΔS°), and Δ(ΔG°)) calculated by the van ’t Hoff method are summarized in Appendix A

The isoenantioselective temperatures (T_iso_) were also calculated as, at this temperature, enthalpy and entropy compensations negate each other, resulting in the coelution of the two enantiomers and no separation. Above and below T_iso_, the EEO is reversed. The Q values (Q = Δ(ΔH°)/T × Δ(ΔS°) 298 K) are used for visualizing the relative contributions of enthalpic and entropic terms to the free energy of adsorption. If the Q value is smaller than one, the enantioseparation is controlled by entropy to a higher extent. If the Q value is higher than one, it indicates that the enthalpy controls the enantioseparation, as observed in most cases.

The retention factors (ln k vs. 1/T) exhibited clearly linear van ‘t Hoff plots in all cases with correlation coefficient higher than 0.98. In our study, alongside the enthalpy-controlled enantioseparation, the less common entropy-driven enantioseparation was frequently observed. Although no global trend could be identified, some conclusions can be drawn. Entropy-controlled enantioseparations are more frequent with amylose-type columns compared to cellulose-type columns. On the Chiralcel OD column, only enthalpy-controlled enantioseparations were observed; for other columns, both enthalpy and entropy-controlled separations were detected. It should also be noted that the Q values were near one in several cases, indicating contributions from both enthalpy and entropy to the enantioseparation. High negative Δ(ΔG°) values reflected high enantioselective discrimination. The lowest Δ(ΔG°) value could be found in the case of omeprazole-MeOH-Chiralpak AD system. It could also be found that MeOH was the most consistent mobile phase for significant enantioselectivity across PPIs. In many cases, extremely high T_iso_ values could be calculated. For example, on the Cellulose-2 column using MeOH for omeprazole and rabeprazole, the theoretically calculated T_iso_ value was higher than 2000 °C, but interestingly, in the case of lansoprazole, it was −82 °C. Temperature-dependent EEO reversal was not observed; however, in a few cases, the T_iso_ value was near to the investigated range (e.g., omeprazole-Chiralcel OD-EtOH 40 °C, omeprazole-Lux cellulose-2-2-PrOH 9 °C, rabeprazole-Chiralpak AD-EtOH 41 °C).

### 2.4. Multivariate Statistical Analysis and Selection of Minimal Representative CSP Set

To evaluate the diversity and similarity of chromatographic behavior across 24 chiral chromatographic systems for the enantioseparation of three PPIs, a multivariate statistical analysis was conducted (barplot, histogram, and pair scatter plots of values are depicted in Figure 5a,b). Rs, Q values, and ΔΔG were used to compute a correlation matrix, which was then visualized as a heatmap (Figure 5c). Note that resolution values were “directed” by adding negative signs to those that corresponded to S-R EEO so that the magnitude of separation efficiency was preserved while information was retained on the EEO reversal. Each cell in this matrix represented the Pearson correlation coefficient (r) between system pairs, reflecting the strength of their enantioselective behavior similarity. Grey cells indicated missing or incalculable values due to a lack of resolution. The Rs-based correlation matrix revealed strong intra-group correlations among systems with identical chiral selectors and different alcohols (e.g., all variants of AD or IA), indicating similar enantioselective trends. Conversely, weak or negative correlations between cellulose- and amylose-based systems (e.g., Chiralcel OD vs. Chiralpak AD) pointed to orthogonal selectivity, likely due to differing enantiorecognition mechanisms. Chiralpak AS looked separate from all the columns. Chiralpak AS was the only column in which the carbamate linkage was not directly connected to the aromatic substituent, and it contained chiral carbon in the substituent part.

To further elucidate structural relationships, hierarchical clustering (dendrogram) was employed (Figure 5d). The resulting dendrogram revealed distinct clusters: for example, Chiralpak AS with 2-PrOH and Lux Cellulose-4 with MeOH formed early diverging branches, indicating highly orthogonal separation behavior. Meanwhile, systems like Chiralcel OD, Chiralpak IA, and Lux Cellulose-2 formed tighter clusters, highlighting redundancy within selector families. Principal Component Analysis (PCA) was also performed to identify underlying trends in the separation data (Figure 5e,f). The first two principal components explained 62.2% of the total variance (as seen on the scree plot), confirming that dimensionality reduction was appropriate. The PCA biplot showed that Rs and Q values for omeprazole were highly correlated while ΔΔG descriptors (particularly for lansoprazole and rabeprazole) aligned differently, indicating that they captured distinct aspects of chiral recognition. Based on these findings, a minimal yet representative set of five chiral chromatographic systems was proposed to capture most of the enantioselective diversity while minimizing experimental redundancy; this is summarized in Table 2.

This selected panel ensures the inclusion of both amylose- and cellulose-based selectors, coated and immobilized phases, and a range of alcohol modifiers. These systems spanned distant branches in the dendrogram and occupied non-overlapping regions in the PCA plot, making them ideal for rapid screening and method development. In summary, this multivariate approach facilitated rational CSP selection, enabling high-efficiency enantioseparation with minimized analytical effort.

### 2.5. Machine Learning for Predicting Directed Resolution

To predict the directed resolution based on the chromatographic system and its associated thermodynamic parameters together with molecular descriptors imported from RDKit, we employed machine learning (ML) models. Two algorithms were chosen as appropriate: the Bayesian Regularized Neural Network (BRNN) and Random Forest (RF). The BRNN is a feedforward neural network with a single hidden layer, regularized using Bayesian techniques to prevent overfitting. It balances model complexity and prediction accuracy by applying a probabilistic framework to weight estimation. The RF is an ensemble learning method that builds multiple decision trees using bootstrap samples and averages their predictions. It handles non-linearity and interactions well and is robust to overfitting and noisy data.

Prior to training, data underwent the following preprocessing steps:

(1) Range scaling of numeric features to standardize input.

(2) Missing value imputation using a PCA-based method.

(3) Removal of zero-variance features (uninformative).

(4) Elimination of highly correlated variables to reduce redundancy and multicollinearity.

To assess model performance while minimizing bias, we used 3-fold cross-validation with five repeats. The performance metric was the root mean square error (RMSE). The best-performing models were then used to predict the training dataset. A strong agreement was observed between predicted and actual directed Rs values, indicating the models effectively captured the relationship between chromatographic conditions and resolution behavior (Figure 6).

### 2.6. Use of Solvent Mixtures as Mobile Phases: Hysteresis of Retention and Selectivity

The use of eluent mixtures in PO mode displayed some advantages; for example, the peak shape, selectivity, and resolution could be altered or fine-tuned in these cases. Shorter analysis times, and even EEO reversal, could be achieved in some cases. In our study, MeOH-EtOH and MeOH-2-PrOH mixtures were investigated on the three amylose-based CSPs.

Interestingly, in our cases, only the CSPs containing amylose (3,5-dimethylphenylcarbamate) showed improvements in enantioseparation using the abovementioned solvent mixtures as eluents. It is worth noting that while, on Chiralpak IA, increased *R*_s_ was observed only for rabeprazole, on the Chiralpak AD column, higher *R*_s_ was observed for all analytes in eluent mixtures. In MeOH-EtOH mixtures using the Chiralpak AS column, there was no observable increase in *R*_s_.

The study of hysteresis revealed that all investigated amylose-based CSPs exhibited this phenomenon. Both retention and selectivity hysteresis were observed (Figure 7), along with a change in the eluent composition at which the reversal of EEO occurred. All hysteresis measurements began from 100 *V*/*V*% MeOH. Interestingly, on Chiralpak AS in MeOH-EtOH mixtures, *R*_s_ values decreased starting from EtOH for all three analytes; however, on Chiralpak AD, applying the same eluent mixture, the same behavior was only observed for omeprazole. Hysteresis was detectable in both eluent systems and corresponding chromatograms are depicted in Figure 8. In MeOH-EtOH mixtures, an interesting phenomenon was observable when the k values of an enantiomer were plotted against the MeOH *V*/*V*%; with the increase in MeOH in the mixture, the lines going forward and backward were diverging (Figure 7). This was not observable in MeOH-2-PrOH mixtures. However, this effect was more pronounced on the Chiralpak AS column than on Chiralpak AD.

In their recent studies, Horváth et al. [25] and Németi et al. [26] introduced a new chromatographic parameter, called hystereticity (υ), which was defined as follows:υ = k_Aforward_/k_Areverse_,(1)

Here, *k*_Aforward_ is the retention factor of the ‘A’ enantiomer measured in the “forward” direction (starting from 100 *V*/*V*% MeOH) while k_Areverse_ is the retention factor measured in the opposite, “reverse” direction (starting from 0 *V*/*V*% MeOH). Németi et al. [26] presented both the log*v* and |*v*−| plots as a function of the mobile phase composition. While these representations have several practical applications, in chiral HPLC, the ratio of the two enantiomers’ hystereticity values can also be plotted as a function of the mobile phase composition, allowing for further observations. Depending on the result of the forward measurement at a given mobile phase composition, two main cases can be identified when examining the hystereticity values:(A) The two enantiomers separate (let k_A_ < k_B_, accordingly EEO A > B);(B) Coelution occurs (k_A_ = k_B_).

For case (A), the reverse measurement can yield three different outcomes:υA = υB, meaning α_forward_ = α_reverse_.υ_A_ > υ_B_, i.e., υ_A_/υ_B_ > 1, which can only occur if k_Areverse_ < k_Breverse_ and either k_Aforward_ > k_Areverse_ or k_Bforward_ < k_Breverse_. This means that either enantiomer A elutes earlier or enantiomer B elutes later, leading to an increase in selectivity.υ_A_ < υ_B_, i.e., υ_A_/υ_B_ < 1, which can only occur if k_Areverse_ ≥ k_Breverse_. If equality holds, the two enantiomers coelute, whereas if k_Areverse_ > k_Breverse_, an EEO reversal occurs.

For case (B), three different outcomes are also possible depending on the result of the reverse measurement:υ_A_ = υ_B_, meaning α_forward_ = α_reverse_ = 0.υ_A_ > υ_B_, i.e., υ_A_/υ_B_ > 1, which can only happen if k_Areverse_ < k_Breverse_, meaning the elution order is A > B.υ_A_ < υ_B_, i.e., υ_A_/υ_B_ < 1, which can only occur if k_Areverse_ > k_Breverse_, meaning the elution order is B > A.

Thus, by examining the result of the forward measurement and the hystereticity values of the two enantiomers at a given mobile phase composition, it can be determined whether an elution order reversal has occurred. Additionally, in the case of forward coelution, if enantiomer separation occurs in the reverse direction, the elution order can be identified using the hysteresis ratio of the two enantiomers.

In our measurements, there were several examples when the ratio of the two enantiomers’ hystereticity presented the same as we experienced in our measurements. We calculated the υ_R_/υ_S_ ratio regardless of the result of the forward measurement. This meant that in the case of the S > R EEO, the reciprocal of our calculated ratio had to be compared to 1. For example, rabeprazole using Chiralpak AD in MeOH-2-PrOH mixtures at 30 *V*/*V*% MeOH had a υ_R_/υ_S_ > 1; however, the forward-going measurement’s EEO was S > R, meaning that we expected at least a coelution or a changed EEO in the backward-going measurement, and the EEO was changed to R > S. A similar result was observed on Chiralpak AS for omeprazole in MeOH-2-PrOH mixtures at 70 *V*/*V*% MeOH. The forward-going measurement had an EEO of S > R, the υ_R_/υ_S_ > 1, and the backward-going measurement resulted in coelution. On Chiralpak IA at 60 *V*/*V*% MeOH, for omeprazole, the forward-going measurement resulted in coelution while the backward resulted in an S > R EEO. Thus, the ratio of hystereticities had to be smaller than 1, as depicted in Figure 7D. In MeOH-EtOH mixtures, there were also examples for the use of the hystereticity ratio. Using the Chiralpak AD column and 70 *V*/*V*% MeOH rabeprazole enantiomers, we found that they had a hystereticity ratio less than 1. The forward-going measurement resulted in an R > S EEO and the backward-going measurement resulted in coelution. Using the same CSP and 90 *V*/*V*% MeOH lasoprazole enantiomers showed the same. *v*_R_/*v*_S_ < 1; the forward-going measurement resulted in an R > S EEO and the backward measurement showed no enantiorecognition. On the Chiralpak AS column, a higher MeOH percentage in the mobile phase resulted in an S > R EEO; thus, the reciprocal of υ_R_/υ_S_ had to be compared to 1. Rabeprazole enantiomers at 40 *V*/*V*% MeOH had a hystereticity ratio greater than 1; however, the forward going measurement resulted in an S > R EEO, while during the backward-going measurements, coelution was obtained (Figure 7H). Similarly, lansoprazole at 80 and 70 *V*/*V*% MeOH displayed an S > R EEO during the forward-going measurements while the backward-going measurements resulted in no enantiorecognition. To summarize, all three compounds exhibited hysteresis on all four amylose-based CSPs in both MeOH–2-PrOH and MeOH–EtOH mobile phase systems. Not only the absolute hystereticity values but also the ratio of hystereticity between the enantiomers provided valuable insights into the separation behavior.

### 2.7. Semi-Preparative Separations

A semi-preparative method was used to collect the distomers of all three analytes. The distomers are not commercially available; however, they are required for several biological studies, such as for protein binding investigations [29]. Full method optimization was not performed; however, small adjustments were made from the screening parameters. The highest *R*_s_ values were chosen for each compound, and the temperature and flow rate were adjusted to gain better separation and shorter analysis times while being aware of the maximum backpressure allowed by the columns’ manufacturer. The methods for these separations were as follows. R-omeprazole was collected on the semi-preparative Chiralpak AD column using neat MeOH with a 40 °C column temperature and 1.5 mL/min flow rate. S-rabeprazole and S-lansoprazole were both collected using the same system on the Lux Cellulose-4 column thermostated at 10 °C using a 1.2 mL/min flow rate. A total of 30 mg of racemic omeprazole was injected, and we gained 13.2 mg of *R*-omeprazole, which was an 88% extraction rate. Quantities of 20 mg of racemic rabeprazole and lansoprazole were separated, and we gained 7.4 mg and 4.3 mg of the distomers, respectively. The extraction rates and purity of the collected enantiomers are summarized in Table 3.

## 3. Materials and Methods

### 3.1. Materials

Omeprazole, lansoprazole, rabeprazole, and S-omeprazole (esomeprazole) were obtained from Sigma-Aldrich, Hungary (Budapest, Hungary). Gradient-grade eluents, including MeOH, EtOH, 1-PrOH, and 2-PrOH, were purchased from Merck (Darmstadt, Germany). R-lansoprazole (dexlansoprazole) and R-rabeprazole (dexrabeprazole) were sourced from Beijing Mesochem Technology (Beijing, China).

Chiralcel OD (250 × 4.6 mm; particle size 10 µm) [based on cellulose tris(3,5-dimethylphenylcarbamate)], Chiralcel OJ (250 × 4.6 mm; particle size 10 µm) [based on cellulose tris(4-methylbenzoate)], Chiralpak AD (250 × 4.6 mm; particle size 10 µm and 250 × 10 mm; particle size 10 µm) [based on amylose tris(3,5-dimethylphenylcarbamate)], Chiralpak AS (250 × 4.6 mm; particle size 10 µm) [based on amylose tris((S)-α-methylbenzyl carbamate)], and Chiralpak IA (250 × 4.6 mm; particle size 10 µm) [based on amylose tris(3,5-dimethylphenylcarbamate]) were products of Daicel Corporation (Tokyo, Japan). Lux Cellulose-2 (150 × 4.6 mm; particle size: 5 µm) [based on cellulose tris(3-chloro-4-methylphenylcarbamate)] and Lux Cellulose-4 (150 × 4.6 mm; particle size: 5 µm) [based on cellulose tris(4-chloro-3-methylphenylcarbamate)] were the products of Phenomenex (Torrance, CA, USA).

### 3.2. HPLC Instrumentation and Conditions

LC-UV analyses were conducted on an Agilent 1100 HPLC instrument comprising an inline degasser (G1322A), a quaternary pump (G1311A), an automatic injector (G1329A) equipped with a sample thermostat (G1330A), a column thermostat (G1316A), and a diode array detector (G1315A), controlled by Agilent Chemstation B04.03-SP2 software. In the screening experiments, a uniform flow rate of 0.7 mL/min was used; for the hysteresis and thermodynamic study, the flow rate was 0.5 mL/min. Column temperature was maintained at 20 °C unless otherwise specified. During the hysteresis study, the eluent composition was changed starting from 100 *V*/*V*% of MeOH, using 10 *V*/*V*% increments until reaching 100 *V*/*V*% of the other solvent and then 10 *V*/*V*% decrements until reaching 100 V/V% MeOH again. A period of 60 min conditioning was applied for each new eluent before first injection. The detection wavelength was 210 nm. The retention factor (k) was determined as k = (t_R_ − t_0_)/t_0_, where t_R_ was the retention time for the eluted enantiomer and t_0_ was the dead time, measured by the interference of MeOH in the baseline. The separation factor (α) was calculated as α = k_2_/k_1_; k_1_ and k_2_ were the retention factors of the first- and second-eluted enantiomers, respectively. R_s_ was calculated with the following formula: R_s_ = 2(t_2_ − t_1_)/(w_1_ + w_2_), where t_1_ and t_2_ were the retention times and w_1_ and w_2_ were the extrapolated peak widths at the baseline for the first and second eluting enantiomers. The impact of column temperature on chromatographic separation was examined at temperatures ranging from 10 to 40 °C. To gain a deeper understanding of the energetic interactions involved, a classical van ‘t Hoff analysis was conducted. This involved constructing van ‘t Hoff plots, where the natural logarithm of the retention factor was plotted against the inverse of the absolute temperature:k = (ΔH°)/RT + ΔS/R + lnΦ,(2)
Here, R stands for the universal gas constant, T is the temperature in Kelvin, and k is the retention factor of the individual enantiomers. ΔH° denotes the standard enthalpy, while ΔS° is the standard entropy change of transfer of the solute from the mobile phase to the stationary phase, and Φ is the phase ratio. Indeed, if ΔH° remains constant within the specified temperature range, plotting the natural logarithm of the retention factor (lnk) against the inverse of the absolute temperature (1/T) typically yields a linear relationship. The slope of this line corresponds to −ΔH°/R (where R is the gas constant) while the intercept is given by ΔS°/R + lnΦ. However, since the phase ratio (Φ) is often unknown, ΔS°* values are frequently employed (where ΔS°* = ΔS° + RlnΦ) to address the uncertainty associated with Φ.

Similarly, the differences in change in standard enthalpy Δ(ΔH°) and standard entropy Δ(ΔS°) for the two enantiomers moving from the mobile phase to the stationary phase were also calculated according to modified van ‘t Hoff equation:lnα = −Δ(ΔH°)/RT + (Δ(ΔS°))/R.(3)

The isoenantioselective temperatures (T_iso_) were determined by calculating the ratio between Δ(ΔH°) and Δ(ΔS°) as follows:T_iso_ = (∆(∆H°))/(∆(∆S°)).(4)

T_iso_ represents the temperature at which the enthalpy contribution is precisely balanced by the entropic term, resulting in a Gibbs free energy (Δ(ΔG°)) value of zero.∆(∆G°) =∆(∆H°) − T*∆(∆S°).(5)

The thermodynamic parameter Δ(ΔG°) provides valuable insights into the binding strength between the analyte and selector in chromatographic separations. More negative values indicate stronger binding interactions, suggesting a more effective separation. When Δ(ΔG°) equals zero, it indicates that there is no difference in binding strength between the enantiomers. Consequently, at the temperature where Δ(ΔG°) is zero (often referred to as the T_iso_), the two enantiomers co-elute and no separation is achieved. Above and below the T_iso_, the enantiomeric elution order typically differs, leading to successful separation.

### 3.3. Multivariate Analysis

#### 3.3.1. Data Preparation

Experimental data describing the chiral separation of three proton pump inhibitors were imported and processed using base R (v 4.4.1) in R Studio (Posit, Boston, MA USA, v 2025.05.0+496). A unique identifier for each chromatographic system was created by combining the chiral stationary phase (CSP) and mobile phase. A new variable, directed resolution (dirRs), was computed by assigning a sign to the resolution (Rs) value based on the enantiomer elution order (EEO), where negative values corresponded to SR elution. Data manipulation and reshaping were performed with dplyr and reshape2 [30,31].

#### 3.3.2. Molecular Descriptor Extraction

Molecular descriptors for omeprazole, rabeprazole, and lansoprazole were extracted from PubChem using the PubChemR package 2.1.4 [32], which provided physicochemical properties such as molecular weight, XLogP, and topological polar surface area. Additional structural and group-based descriptors were calculated using ChemmineR (https://www.bioconductor.org/packages/devel/bioc/vignettes/ChemmineR/inst/doc/ChemmineR.html access on 22 July 2025)) [33], while atomic, bond, and fingerprint descriptors were generated using Rcpi (https://www.bioconductor.org/packages/devel/bioc/vignettes/Rcpi/inst/doc/Rcpi.html#Introduction (access data on 22 July 2025)) [34]. These descriptors were merged with the experimental data for use in downstream modeling.

#### 3.3.3. Visualization and Correlation Analysis

Descriptive plots were created using ggplot2 [35] and multiple plots were arranged with gridExtra and cowplot [36,37]. Resolution (dirRs), thermodynamic (ΔΔG), and selectivity (Q) values were visualized in bar plots faceted by mobile phase. Pairwise Pearson correlation matrices for both dirRs and Q across chromatographic methods were computed and visualized as heatmaps using ggplot2 and reshape2.

#### 3.3.4. Clustering and Principal Component Analysis (PCA)

Hierarchical clustering was conducted on scaled values of ΔΔG, Q, and dirRs weblin the hclust() function with Euclidean distance. The resulting dendrogram was visualized and colored by cluster using dendextend [38]. PCA was performed on the same feature set using FactoMineR [39], and results were visualized with factoextra [40]. A scree plot was used to determine the number of principal components to retain and a biplot with cos^2^-based coloring illustrated variable contributions.

#### 3.3.5. Machine Learning Modeling

To predict directed resolution (dirRs) from system characteristics and molecular descriptors, two supervised machine learning models were employed: Bayesian Regularized Neural Network (BRNN) [41]—a neural network regularized via Bayesian inference to reduce overfitting; Random Forest (RF) [42]—an ensemble method based on multiple decision trees. Prior to modeling, features were preprocessed using caret [43] to apply range scaling, mean imputation for missing values, removal of near-zero variance features, and elimination of highly correlated variables (correlation cutoff = 0.999).

Model training was conducted using 3-fold cross-validation repeated 5 times to reduce bias and variance in performance estimation. Hyperparameter tuning was carried out via grid search. Parallel processing was enabled using doParallel [44] to accelerate computation. Models were evaluated using root mean square error (RMSE) and predictions were plotted against true values, showing high correlation and minimal deviation from the line of equality.

## 4. Conclusions

In this study, the enantiomeric separation of three proton pump inhibitors (omeprazole, lansoprazole, and rabeprazole) was systematically investigated using seven polysaccharide-based chiral stationary phases in a PO mode. The use of different neat alcohols and their mixtures as eluents allowed for the optimization of separation conditions, with methanol emerging as the most effective eluent for achieving baseline separation on the Chiralpak AS column. Notably, enantiomer elution order reversal was observed in numerous instances, influenced by both the chiral selector backbone and the choice of eluents. This phenomenon highlights the complex interplay between the stationary phase, eluent composition, and enantiomer interactions. Thermodynamic studies revealed enthalpy- and entropy-controlled enantioseparation processes although temperature-dependent elution order reversal was not observed. Furthermore, hysteresis in retention time, selectivity, and enantiomer elution order was systematically evaluated, demonstrating its presence across all amylose-type columns, regardless of the eluent mixtures used. The calculated hystereticity values provided a quantitative measure to compare hysteresis behavior across different systems. Multivariate statistical analysis, including correlation matrices, hierarchical clustering, and principal component analysis, revealed clear differences in the enantioselective behavior of the investigated systems. Chiralpak AS, for example, showed distinct chromatographic behavior and formed a separate cluster, likely due to its unique selector structure. The statistical analysis also enabled the rational selection of a minimal yet representative set of chiral columns that captured the major diversity of enantioselective trends across the full dataset. Taken together, these results provide valuable insights into the mechanisms of enantiomer recognition and highlight the importance of chromatographic parameter selection. Due to its comprehensive characterization and systematic approach, the methodology presented herein may serve as a solid foundation for the development of pharmacopoeial methods for chiral drugs containing sulfoxide moieties.

## Figures and Tables

**Figure 1 ijms-26-07217-f001:**
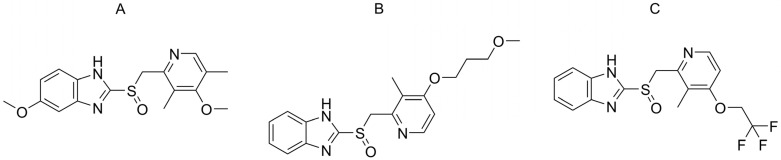
Chemical structure of the investigated chiral sulfoxides ((**A**) omeprazole, (**B**) rabeprazole, (**C**) lansoprazole).

**Figure 2 ijms-26-07217-f002:**
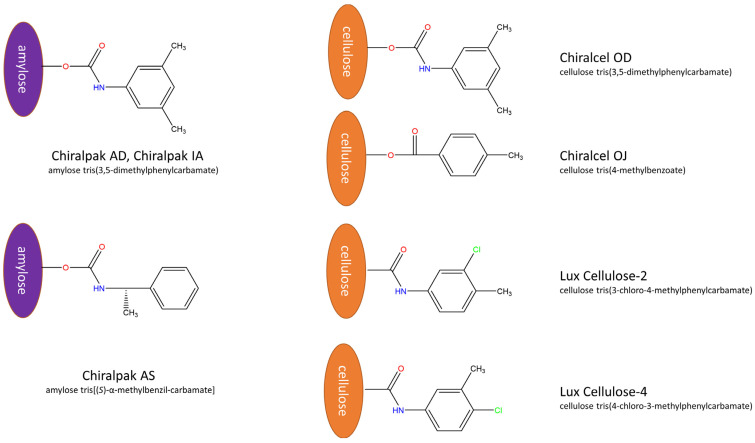
The chemical structures of the chiral selectors used in this study.

**Figure 3 ijms-26-07217-f003:**
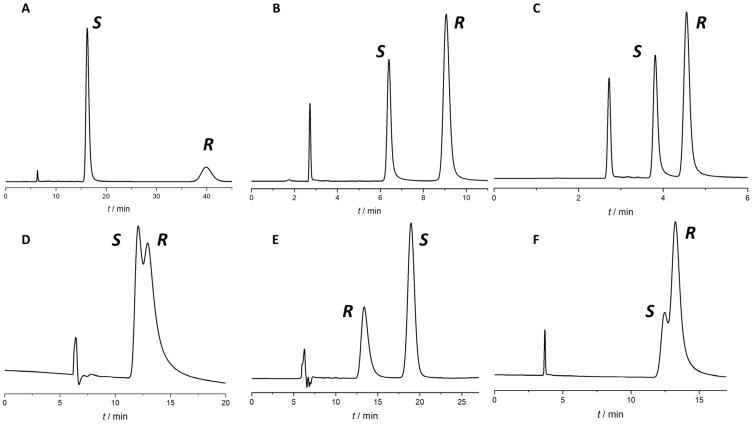
Some representative chromatograms from the screening study. (**A**) Omeprazole on Chiralpak AD with MeOH; (**B**) rabeprazole on Lux Cellulose-4 with MeOH; (**C**) lansoprazole on Lux Cellulose-4 with MeOH; (**D**) omeprazole on Chiralcel OD with 2-PrOH; (**E**) omeprazole on Chiralpak AD with 1-PrOH; (**F**) rabeprazole on Lux Cellulose-2 with EtOH. Flow rate: 0.7 mL/min, temperature: 20 °C.

**Figure 4 ijms-26-07217-f004:**
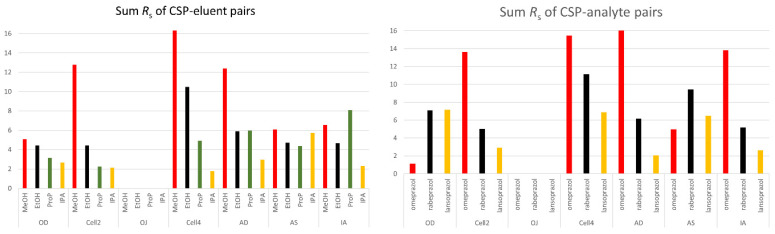
Characterization of the success rate of the separation system with sum Rs values: sum Rs of CSP–eluent pairs (**left**) and sum R_s_ of CSP–analyte pairs (**right**).

**Figure 5 ijms-26-07217-f005:**
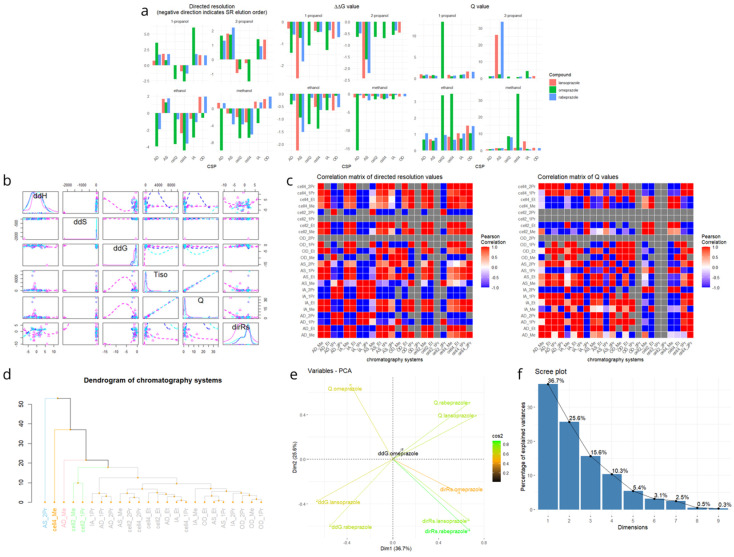
(**a**) Values of directed resolution, ΔΔG, and Q with bar plots in facet view to visualize the effect of CSP and mobile phase on each enantioseparation. Interestingly, the outlier values for ΔΔG and Q belonged to different chromatography systems. (**b**) Pairs scatter plots of the thermodynamic parameters, with the smoothed histogram of each parameter in the diagonal. (**c**) Correlation matrices of the 24 chromatography systems calculated with directed resolution (left) and Q (right). Grey areas denote missing values, where the correlation could not be calculated. (**d**) Dendrogram from hierarchical clustering based on Gibbs free energy (ΔΔG), Q value, and directed resolution (Rs) data. The structure of a dendrogram contains (1) leaves (or tips), which are the individual observations (chromatography methods) being clustered; (2) branches, which are lines connecting observations or clusters and show how data points are grouped step-by-step; (3) nodes, which are points where two branches merge and represents the fusion of two clusters; (4) height (*y*-axis), which reflects the dissimilarity (distance) at which clusters are merged; the higher the node is, the more dissimilar the clusters being joined are. (**e**) Principal Component Analysis (PCA) biplot of chromatography systems as points and variables (Rs, Q, ΔΔG) as vectors, combined with cos^2^ values that indicate the quality of representation of each variable on the principal components. Vectors with similar directions indicate correlated variables; opposite directions imply negative correlation. (**f**) The scree plot of PCA illustrating the proportion of variance explained by each principal component.

**Figure 6 ijms-26-07217-f006:**
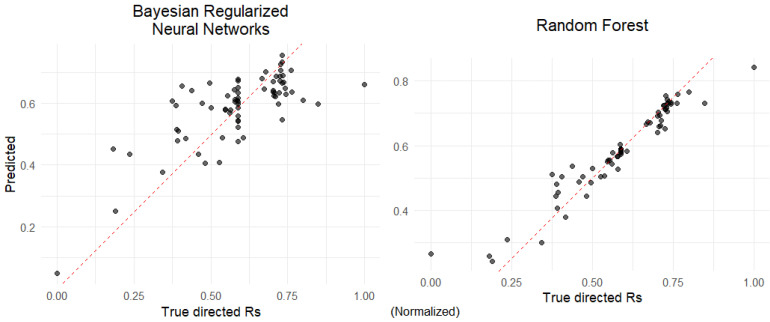
Predicted vs. actual directed resolution values for BRNN (**left**) and RF (**right**). Red dashed line represents perfect correlation (y = x). Note that directed resolution values were normalized with range scaling into the [0, 1] interval.

**Figure 7 ijms-26-07217-f007:**
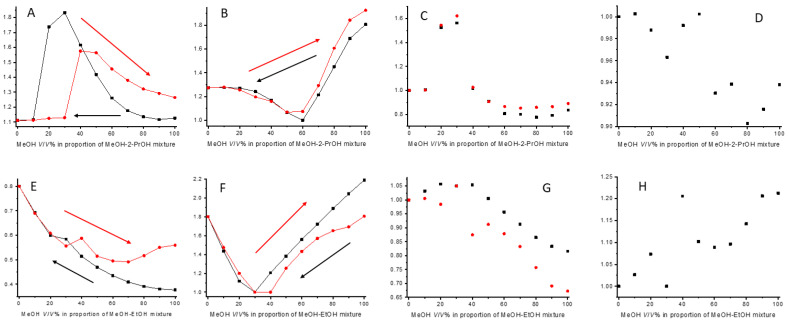
Different chromatographic parameters depicted in *V/V*% of MeOH. (**A**) k_S_ of omperazole, Chiralpak IA, MeOH-2-PrOH; (**B**) α of omeprazole, Chiralpak IA, MeOH-2-PrOH; (**C**) v_R_ (black) and v_S_ (red) of OME, Chiralpak IA, MeOH-2-PrOH; (**D**) v_R_/v_S_ of omeprazole, Chiralpak IA, MeOH-2-PrOH; (**E**) k_S_ of rabeprazole, Chiralpak AS, MeOH-EtOH; (**F**) α of rabeprazole, Chiralpak AS, MeOH-EtOH; (**G**) v_R_ (black) and v_S_ (red) of rabeprazole, Chiralpak AS, MeOH-EtOH; (**H**) v_R_/v_S_ of rabeprazole, Chiralpak AS, MeOH-EtOH.

**Figure 8 ijms-26-07217-f008:**
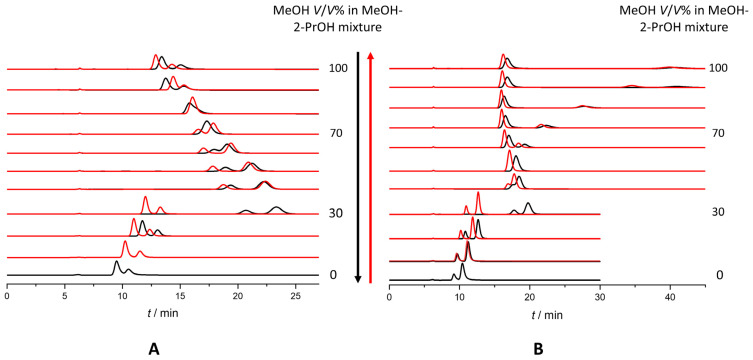
Chromatograms of hysteresis. Black: going from 100 *V*/*V*% MeOH and red: going from 0 *V*/*V*% MeOH. (**A**) Rabeprazole, Chiralpak AD, MeOH-2-PrOH; (**B**) omeprazole, Chiralpak AD, MeOH-2-PrOH.

**Table 1 ijms-26-07217-t001:** Results of the screening (k_1_ and k_2_ are the retention factors of the enantiomers, R_s_ is the resolution, and EEO is the enantiomeric elution order).

		Omeprazol				Rabeprazol				Lansoprazol			
Column	Eluent	k_1_	k_2_	R_s_	EEO	k_1_	k_2_	R_s_	EEO	k_1_	k_2_	R_s_	EEO
OD	MeOH	0.20	0.20	-	-	0.23	0.43	2.83	R-S	0.04	0.16	2.25	R-S
	EtOH	0.42	0.47	0.57	S-R	0.51	0.69	1.95	R-S	0.33	0.48	1.92	R-S
	1-PrOH	0.43	0.43	-	-	0.48	0.69	1.55	R-S	0.39	0.58	1.60	R-S
	2-PrOH	0.88	0.88	-	-	1.17	1.17	-	-	0.87	1.24	1.38	R-S
Cell2	MeOH	1.02	1.90	7.02	S-R	1.54	2.11	3.76	S-R	0.53	0.68	2.00	S-R
	EtOH	1.32	2.18	3.69	S-R	2.39	2.60	0.74	S-R	0.93	0.93	-	-
	1-PrOH	1.54	2.41	2.24	S-R	2.90	2.90	-	-	1.32	1.32	-	-
	2-PrOH	3.77	4.89	0.68	S-R	6.69	6.69	-	-	2.81	3.65	0.94	S-R
OJ	MeOH	0.23	0.23	-	-	0.18	0.18	-	-	0.07	0.07	-	-
	EtOH	0.22	0.22	-	-	0.15	0.15	-	-	0.05	0.05	-	-
	1-PrOH	0.27	0.27	-	-	0.19	0.19	-	-	0.13	0.13	-	-
	2-PrOH	0.41	0.41	-	-	0.31	0.31	-	-	0.23	0.23	-	-
Cell4	MeOH	0.85	1.79	6.89	S-R	1.35	2.33	6.16	S-R	0.40	0.67	3.26	S-R
	EtOH	1.27	3.03	4.44	S-R	2.56	4.11	3.62	S-R	0.82	1.33	2.41	S-R
	1-PrOH	1.45	2.45	2.60	S-R	3.21	3.97	1.36	S-R	1.08	1.37	0.97	S-R
	2-PrOH	3.87	5.69	1.52	S-R	6.54	6.54	-	-	3.30	3.73	0.26	S-R
AD	MeOH	1.77	5.88	9.91	S-R	1.19	1.41	1.27	R-S	0.53	0.79	1.33	R-S
	EtOH	1.82	2.65	3.95	S-R	1.80	2.14	1.93	S-R	1.02	1.02	-	-
	1-PrOH	1.13	2.02	3.59	R-S	0.913	1.16	1.67	R-S	0.59	0.67	0.72	R-S
	2-PrOH	0.53	0.75	1.65	R-S	0.573	0.76	1.30	R-S	0.35	0.35	-	-
AS	MeOH	0.41	0.54	1.18	S-R	0.38	0.85	3.72	S-R	0.14	0.24	1.19	S-R
	EtOH	0.52	0.73	1.27	R-S	0.41	0.69	1.75	R-S	0.15	0.36	1.68	R-S
	1-PrOH	0.52	0.72	0.80	R-S	0.33	0.71	1.76	R-S	0.19	0.53	1.82	R-S
	2-PrOH	1.15	2.18	1.73	R-S	0.76	1.77	2.22	R-S	0.43	1.08	1.80	R-S
IA	MeOH	1.32	2.26	3.45	S-R	0.78	0.90	1.44	R-S	0.35	0.46	1.65	R-S
	EtOH	1.39	1.84	2.92	S-R	1.29	1.42	1.08	S-R	0.64	0.69	0.68	S-R
	1-PrOH	0.78	1.55	6.02	R-S	0.87	1.07	1.75	R-S	0.56	0.56	-	-
	2-PrOH	0.87	1.09	1.41	R-S	0.99	1.15	0.91	R-S	0.64	0.64	-	-

**Table 2 ijms-26-07217-t002:** Proposed minimal set of five chiral chromatographic systems selected based on PCA of resolution (Rs), selectivity (Q), and ΔΔG values. These systems collectively captured the main enantioselective diversity observed across the analytes while reducing experimental redundancy.

Column	Alcohol	Rationale
Chiralpak AS	2-PrOH	Most distant branch in dendrogram; unique behavior
Cellulose-4	MeOH	Orthogonal to amylose; strong enantioselectivity
Chiralcel OD	1-PrOH	Representative of moderate selectivity group
Chiralpak IA	EtOH	Bridges IA/AD cluster; distinct profile
Lux Cellulose-2	EtOH	Covers Lux Cellulose-2 cluster with non-redundant behavior

**Table 3 ijms-26-07217-t003:** Extraction yield and enantiomeric purity of isolated enantiomers following semi-preparative chiral chromatography. R-omeprazole, S-rabeprazole, and S-lansoprazole were collected under optimized conditions using Chiralpak AD and Lux Cellulose-4 columns.

Collected Enantiomers	Extraction Rate	Purity (%)
R-omeprazole	88%	99.93%
S-rabeprazole	74%	99.89%
S-lansoprazole	43%	99.85%

## Data Availability

The original contributions presented in this study are included in the article/Appendix A; further inquiries can be directed to the corresponding author.

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
