# Peer review of "Enantioseparation of Proton Pump Inhibitors by HPLC on Polysaccharide-Type Stationary Phases: Enantiomer Elution Order Reversal, Thermodynamic Characterization, and Hysteretic Effect"

_ijms, 2025, doi:10.3390/ijms26157217_

Round 1

Reviewer 1 Report

Comments and Suggestions for Authors

The manuscript presented by Dobó and Co-workers shows a comprehensive and systematic study on the enantioseparation of three proton pump inhibitors (PPIs) using polysaccharide-based chiral stationary phases (CSPs) under polar organic (PO) mode conditions. The authors investigate key chromatographic phenomena such as enantiomer elution order (EEO) reversal, thermodynamic behavior, and hysteresis in retention/selectivity. The study is technically sound and methodologically detailed. However, its novelty is marginal, and some redundancies with existing literature limit the impact.

Based on the previous comments, it is recommended to address the following issues:

  1. Although comprehensive, the methods employed, such as polar organic mode using polysaccharide-based chiral stationary phases, and the selected analytes (proton pump inhibitors) have been previously explored. This work offers refinements but does not introduce fundamentally novel concepts. The authors themselves cite several closely related studies (e.g., Cirilli et al., 2008; Papp et al., 2021; Foroughbakhshfasaei et al., 2021), highlighting the limited conceptual novelty. Additionally, the manuscript does not fully adhere to the IJMS reference formatting guidelines. According to the MDPI Vancouver style, all authors must be listed in each reference entry.
  2. It was observed that the manuscript demonstrates redundancy in the cited literature, as its premise significantly overlaps with recent studies addressing hysteresis and enantiomer elution order reversal. The contribution appears marginal, with novelty arising primarily from the systematic combination of variables rather than from any substantive conceptual or methodological innovation.
  3. The exhaustive matrix of CSPs, eluents, and temperatures risks data overload without clear innovation. There is data saturation without innovation, for instance, many of the thermodynamic observations (e.g., linear van’t Hoff plots, ΔΔH°, ΔΔS° calculation) are expected and do not yield unexpected mechanistic findings.
  4. While the paper mentions chiral selector substituents and backbone (cellulose vs. amylose), it lacks deeper structural analysis or molecular modeling to support hypotheses on recognition mechanisms. Here is a recommendation to enrich the discussion of structural influence in the separation.
  5. The manuscript has limitations in the data interpretation, for instance, while the T_iso and ΔΔG° data are presented, their correlation with structural features or predictive models is underdeveloped. Besides, no statistical analysis or chemometric modeling (e.g., PCA, clustering) is used to strengthen mechanistic interpretation.
  6. Although methodologically sound and carefully executed, the originality is limited. The manuscript does not propose new CSPs, new chiral analytes, or novel separation principles. Its added value lies in the exhaustive application of known phenomena (especially hysteresis and EEO reversal) to a well-defined chemical class (PPIs). As such, it is incremental rather than disruptive in its contribution.

If these issues are not adequately addressed, the manuscript may not meet the publication standards of IJMS in its current form. Major revisions are therefore recommended before any consideration for acceptance.

Author Response

The manuscript presented by Dobó and Co-workers shows a comprehensive and systematic study on the enantioseparation of three proton pump inhibitors (PPIs) using polysaccharide-based chiral stationary phases (CSPs) under polar organic (PO) mode conditions. The authors investigate key chromatographic phenomena such as enantiomer elution order (EEO) reversal, thermodynamic behavior, and hysteresis in retention/selectivity. The study is technically sound and methodologically detailed. However, its novelty is marginal, and some redundancies with existing literature limit the impact.

Based on the previous comments, it is recommended to address the following issues:

  1. Although comprehensive, the methods employed, such as polar organic mode using polysaccharide-based chiral stationary phases, and the selected analytes (proton pump inhibitors) have been previously explored. This work offers refinements but does not introduce fundamentally novel concepts. The authors themselves cite several closely related studies (e.g., Cirilli et al., 2008; Papp et al., 2021; Foroughbakhshfasaei et al., 2021), highlighting the limited conceptual novelty. Additionally, the manuscript does not fully adhere to the IJMS reference formatting guidelines. According to the MDPI Vancouver style, all authors must be listed in each reference entry. It was observed that the manuscript demonstrates redundancy in the cited literature, as its premise significantly overlaps with recent studies addressing hysteresis and enantiomer elution order reversal. The contribution appears marginal, with novelty arising primarily from the systematic combination of variables rather than from any substantive conceptual or methodological innovation.

We appreciate your opinion; however, we believe that our manuscript contains sufficient novelty to merit publication. In response to the reviewers’ comments and to further emphasize the originality of our work, we have revised the manuscript accordingly. Specifically, we have incorporated a novel statistical analysis of the thermodynamic data, supplemented by machine learning techniques. Additionally, we have clarified the novelty more explicitly in the revised Introduction and Conclusion sections. The reference list has been expanded and corrected in the revised version of the manuscript. The following points represent the key novel aspects of our manuscript:

  • A systematic and comprehensive evaluation, including detailed investigation of hysteresis, thermodynamic behavior, and particularly enantiomer elution order reversal, which provides significant methodological insights and contributes to a deeper understanding of the underlying chromatographic mechanisms.
  • A novel and simple method for the enantioseparation of various proton pump inhibitors, which may serve as a foundation for future pharmacopeial methods.
  • A semi-preparative scale enantioseparation approach for proton pump inhibitors under polar organic mode conditions, offering a practical route for the preparation of enantiopure reference standards.
  • A comprehensive statistical analysis, including the application of machine learning algorithms, to investigate the thermodynamic parameters and enantioselective recognition mechanisms on polysaccharide-based chiral stationary phases in polar organic mode.
  • The introduction of a novel and meaningful calculation method for quantifying hysteresis effects, which may assist future chromatographic studies involving complex retention behaviors.
  1. The exhaustive matrix of CSPs, eluents, and temperatures risks data overload without clear innovation. There is data saturation without innovation, for instance, many of the thermodynamic observations (e.g., linear van’t Hoff plots, ΔΔH°, ΔΔS° calculation) are expected and do not yield unexpected mechanistic findings. The manuscript has limitations in the data interpretation, for instance, while the Tiso and ΔΔG° data are presented, their correlation with structural features or predictive models is underdeveloped. Besides, no statistical analysis or chemometric modeling (e.g., PCA, clustering) is used to strengthen mechanistic interpretation.

We highly appreciate your comments. We add to the following section to the manuscript.

2.4. Multivariate Statistical Analysis and Selection of Minimal Representative CSP Set

To evaluate the diversity and similarity of chromatographic behavior across 24 chiral chromatographic systems for the enantioseparation of three PPIs, a multivariate statistical analysis was conducted (barplot, histogram and pair scatter plots of values are depicted in Figure 5a-b). Rs, Q values, and ΔΔG were used to compute a correlation matrix, which was then visualized as a heatmap (Figure 5c). Note that resolution values were “directed” by adding negative sign to those which correspond to S-R EEO, so that the magnitude of separation efficiency is preserved while information is retained on the EEO reversal. Each cell in this matrix represented the Pearson correlation coefficient (r) between system pairs, reflecting the strength of their enantioselective behavior similarity. Grey cells indicated missing or incalculable values due to lack of resolution. The Rs-based correlation matrix revealed strong intra-group correlations among systems with identical chiral selectors and different alcohols (e.g., all variants of AD or IA), indicating similar enantioselective trends. Conversely, weak or negative correlations between cellulose- and amylose-based systems (e.g., Chiralcel OD vs. Chiralpak AD) pointed to orthogonal selectivity, likely due to differing enantiorecognition mechanisms. Chiralpak AS looks separate from all the columns. Chiralpak AS is the only column in which the carbamate linkage is not directly connected to the aromatic substituent, and contain chiral carbon in the substituent part.

To further elucidate structural relationships, hierarchical clustering (dendrogram) was employed (Figure 5d). The resulting dendrogram revealed distinct clusters: for example, Chiralpak AS with 2-PrOH and Lux Cellulose-4 with MeOH formed early diverging branches, indicating highly orthogonal separation behavior. Meanwhile, systems like Chiralcel OD, Chiralpak IA, and Lux Cellulose-2 formed tighter clusters, highlighting redundancy within selector families. Principal component analysis (PCA) was also performed to identify underlying trends in the separation data (Figure 5e-f). The first two principal components explained 62.2% of the total variance (as seen on the scree plot), confirming that dimensionality reduction was appropriate. The PCA biplot showed that Rs and Q values for omeprazole were highly correlated, while ΔΔG descriptors (particularly for lansoprazole and rabeprazole) aligned differently, indicating they capture distinct aspects of chiral recognition. Based on these findings, a minimal yet representative set of five chiral chromatographic systems was proposed to capture most of the enantioselective diversity while minimizing experimental redundancy, that is summarized in Table 2.

Table 2. Proposed minimal set of five chiral chromatographic systems selected based on PCA of resolution (Rs), selectivity (Q), and ΔΔG values. These systems collectively capture the main enantioselective diversity observed across the analytes while reducing experimental redundancy.

Column

Alcohol

Rationale

Chiralpak AS

2-PrOH

Most distant branch in dendrogram; unique behavior

Cellulose-4

MeOH

Orthogonal to amylose; strong enantioselectivity

Chiralcel OD

1-PrOH

Representative of moderate selectivity group

Chiralpak IA

EtOH

Bridges IA/AD cluster; distinct profile

Lux Cellulose-2

EtOH

Covers Lux Cellulose-2 cluster with non-redundant behavior

This selected panel ensures the inclusion of both amylose and cellulose-based selectors, coated and immobilized phases, and a range of alcohol modifiers. These systems span distant branches in the dendrogram and occupy non-overlapping regions in the PCA plot, making them ideal for rapid screening and method development. In summary, this multivariate approach facilitates rational CSP selection, enabling high efficiency enantioseparation with minimized analytical effort.

Figure 5. (a) Values of directed resolution, ΔΔG, and Q with bar plots in facet view to visualize the effect of CSP and mobile phase on each enantioseparation. Interestingly, the outlier values for ΔΔG and Q belong to different chromatography systems. (b) Pairs scatter plots of the thermodynamic parameters, with the smoothed histogram of each parameter in the diagonal. (c) Correlation matrices of the 24 chromatography systems calculated with directed resolution (left) and Q (right). Grey areas denote missing values, where the correlation could not be calculated. (d) Dendrogram from hierarchical clustering based on Gibbs free energy (ΔΔG), Q value, and directed resolution (Rs) data. The structure of a dendrogram contains: (1) leaves (or tips) which are the individual observations (chromatography methods) being clustered; (2) branches, which are lines connecting observations or clusters and show how data points are grouped step-by-step; (3) nodes are points where two branches merge and represents the fusion of two clusters; (4) height (y-axis) reflects the dissimilarity (distance) at which clusters are merged; the higher the node, the more dissimilar the clusters being joined. (e) Principal Component Analysis (PCA) biplot of chromatography systems as points and variables (Rs, Q, ΔΔG) as vectors, combined with cos² values that indicate the quality of representation of each variable on the principal components. Vectors with similar directions indicate correlated variables; opposite directions imply negative correlation. (f) The scree plot of PCA illustrating the proportion of variance explained by each principal component.

2.5. Machine Learning for Predicting Directed Resolution

To predict the directed resolution based on the chromatographic system and its associated thermodynamic parameters together with molecular descriptors imported from RDKit, we employed machine learning (ML) models. Two algorithms were chosen as appropriate: Bayesian Regularized Neural Networks (BRNN) and Random Forests (RF). BRNN is a feedforward neural network with a single hidden layer, regularized using Bayesian techniques to prevent overfitting. It balances model complexity and prediction accuracy by applying a probabilistic framework to weight estimation. RF is an ensemble learning method that builds multiple decision trees using bootstrap samples and averages their predictions. It handles non-linearity and interactions well, and is robust to overfitting and noisy data.

Prior to training, data underwent the following preprocessing steps: (1) Range scaling of numeric features to standardize input. (2) Missing value imputation using a PCA-based method. (3) Removal of zero-variance features (uninformative). (4) Elimination of highly correlated variables to reduce redundancy and multicollinearity. To assess model performance while minimizing bias, we used 3-fold cross-validation with 5 repeats. The performance metric was the root mean square error (RMSE). The best-performing models were then used to predict the training dataset. A strong agreement was observed between predicted and actual directed Rs values, indicating the models effectively captured the relationship between chromatographic conditions and resolution behavior (Figure 6).

Figure 6. Predicted vs. actual directed resolution values for BRNN (left) and RF (right). Red dashed line represents perfect correlation (y = x). Note that directed resolution values are normalized with range scaling into the [0, 1] interval.

  1. While the paper mentions chiral selector substituents and backbone (cellulose vs. amylose), it lacks deeper structural analysis or molecular modeling to support hypotheses on recognition mechanisms. Here is a recommendation to enrich the discussion of structural influence in the separation.

Thank you for your remarks. The manuscript has been supplemented with a more in-depth structural analysis regarding the enantiorecognition mechanism. (See the previous comments as well)

Reviewer 2 Report

Comments and Suggestions for Authors

Article: Enantioseparation of proton pump inhibitors by HPLC on polysaccharide-type stationary phases: enantiomer elution order reversal, thermodynamic characterization and hysteretic effect

Authors: Máté Dobó, Gergely Molnár, Ali Mhammad, Gergely Dombi, Zoltán-István Szabó and GergÅ‘ Tóth

The authors performed 84 measurements covering 7 stationary phases of the polysaccharide type: with amylose or cellulose backbone, 4 pure eluents (methanol, ethanol, 1-propanol, 2-propanol) and 3 IPP: omeprazole, rabeprazole, lansoprazole. Enantiomeric separation was achieved with an efficiency of 70% and, while satisfactory separation to the baseline was achieved in 38.5%. The optimization of the enantiomer elution order reversal (EEO), thermodynamic aspects of enantiomeric separation and, retention and selectivity hysteresis were discussed.

The work is written in an understandable language with a clearly defined goal that has been achieved. Nevertheless, I would be grateful if the Authors could answer the following questions.

Line 80, page 2; The abbreviation PO should be entered in the text the first time it appears.

Line 113, page 3 eluent, eluents in neat form or pure eluents, rather than neat eluents.

Could the authors try to explain why there was no PPIs separations on the Chiralcel OJ column?

Which form, S or R, for individual PPIs is the active form and therefore such an elution order is desirable, where the enantiomer devoid of biological activity elutes first. This is particularly important for those PPIs that have been registered as enantiomerically pure forms.

Can the conclusions drawn by the authors be applied more generally to other chiral sulfur compounds, or only to PPIs?

In Figure 5, the x and y coordinate axes are illegible. Please solve the problem.

Line 319, page 10; EEO was introduced for elution order, although this abbreviation had appeared before.

Line 365, page 12; What about samples that were separated preparatively – what was the purpose of this step? Was a polarimetric measurement performed later to confirm the purity of the separated enantiomers?

Author Response

The authors performed 84 measurements covering 7 stationary phases of the polysaccharide type: with amylose or cellulose backbone, 4 pure eluents (methanol, ethanol, 1-propanol, 2-propanol) and 3 IPP: omeprazole, rabeprazole, lansoprazole. Enantiomeric separation was achieved with an efficiency of 70% and, while satisfactory separation to the baseline was achieved in 38.5%. The optimization of the enantiomer elution order reversal (EEO), thermodynamic aspects of enantiomeric separation and, retention and selectivity hysteresis were discussed. The work is written in an understandable language with a clearly defined goal that has been achieved. Nevertheless, I would be grateful if the Authors could answer the following questions.

We highly appreciate your positive feedback. We have modified our revised manuscript according to reviewers' comments and have taken all remarks into consideration.

Line 80, page 2; The abbreviation PO should be entered in the text the first time it appears.

Corrected.

Line 113, page 3 eluent, eluents in neat form or pure eluents, rather than neat eluents.

Corrected.

Could the authors try to explain why there was no PPIs separations on the Chiralcel OJ column?

Chiralcel OJ is the only column that utilizes a benzoate-type selector, while all other columns feature a carbamate linker. The carbamate group may play a significant role in the enantiorecognition mechanisms.

Which form, S or R, for individual PPIs is the active form and therefore such an elution order is desirable, where the enantiomer devoid of biological activity elutes first. This is particularly important for those PPIs that have been registered as enantiomerically pure forms.

We thank the reviewer for this insightful question. As discussed in the Introduction, several PPIs are currently marketed in enantiomerically pure form: esomeprazole (S-omeprazole), dexlansoprazole (R-lansoprazole), and dexrabeprazole (R-rabeprazole), based on their superior pharmacological profiles compared to their distomers [11,12]. Accordingly, the biologically active enantiomer varies by compound. In our study, we placed emphasis on determining the enantiomer elution order (EEO) under various chromatographic conditions, as this is essential for analytical and preparative purposes. Ideally, for quality control applications, the distomer—typically the less active or inactive enantiomer—should elute first, facilitating its accurate quantification.

Importantly, since the distomers of the investigated PPIs are not commercially available, we performed semi-preparative scale enantioseparations specifically to isolate these enantiomers. This enabled us to obtain highly pure distomer fractions, which are required for further biological studies such as protein binding assays. Thus, the EEO information and the developed methods provide both analytical value and practical utility in research and pharmaceutical development.

Can the conclusions drawn by the authors be applied more generally to other chiral sulfur compounds, or only to PPIs?

We thank the reviewer for this insightful question. Although our study specifically focused on three proton pump inhibitors with chiral sulfoxide moieties, several of the key conclusions can be generalized to other chiral sulfur-containing compounds. First, the chromatographic phenomena we observed—such as eluent-dependent and selector-dependent enantiomer elution order (EEO) reversal, thermodynamically distinct enantioseparation mechanisms (enthalpy- vs. entropy-driven), and hysteresis behavior—are not unique to PPIs. These are governed primarily by the physicochemical interactions between the chiral sulfoxide group and the polysaccharide-based stationary phase, suggesting that structurally related chiral sulfur compounds may exhibit similar behaviors under polar organic conditions. Second, we conducted multivariate statistical analysis (including hierarchical clustering and PCA) to capture and rationalize the chromatographic diversity of the tested systems. This approach allowed us to identify a minimal yet representative set of chiral stationary phases and mobile phase combinations. The underlying statistical structure of the data—such as trends in ΔΔG, Q values, and resolution—reflects general enantioselective behavior patterns that could guide the method development for other chiral sulfoxides beyond proton pump inhibitors.

Moreover, the machine learning models developed in our revised study were trained on molecular descriptors, chromatographic conditions, and thermodynamic data. These models showed high predictive power for directed resolution values, suggesting that similar ML pipelines can be adapted to predict enantioseparation outcomes for other chiral sulfur compounds, provided that relevant molecular descriptors are available. In this context, our work offers both a conceptual framework and practical tools that are extendable to broader compound classes. In summary, while the exact chromatographic behavior is analyte-dependent and would require experimental confirmation, the mechanistic insights, statistical analyses, and predictive modeling strategies described here provide a transferable foundation for studying and optimizing the enantioseparation of other chiral sulfoxides.

In Figure 5, the x and y coordinate axes are illegible. Please solve the problem.

Corrected.

Line 319, page 10; EEO was introduced for elution order, although this abbreviation had appeared before.

Corrected.

Line 365, page 12; What about samples that were separated preparatively – what was the purpose of this step? Was a polarimetric measurement performed later to confirm the purity of the separated enantiomers?

A semi-preparative method was used to collect the distomers of all three analytes. The distomers are not commercially available; however, they are required for several biological studies, such as protein binding investigations. A new table has been added to the revised manuscript, summarizing the extraction rates and purities of all isolated enantiomers.

Round 2

Reviewer 1 Report

Comments and Suggestions for Authors

After reviewing the revised manuscript and confirming that the authors have thoroughly addressed all comments and suggestions, it is recommended that the manuscript be accepted. However, there is a minor issue regarding the size of the graphs presented in Figure 5. The current layout makes them difficult to interpret. It is suggested that the authors rearrange or resize the graphs to improve clarity and readability.